# Transfer Learning and Quantization for Efficient AP vs. LA X-Ray View Classification on an Edge Device

**Keshav Bimbraw**[1, iD]                        KESHAV.BIMBRAW@ONPOINTSURGICAL.COM

**Daniel Steines**[1]                              DANIEL.STEINES@ONPOINTSURGICAL.COM

[1] *OnPoint Surgical, Inc., 555 Virginia Road, Ste 103, Concord, MA 01742, USA*

## Abstract

In this paper, we present a framework for classifying X-ray images as either anterior-posterior (AP) or lateral (LA) by combining transfer learning with model quantization to optimize deep learning models for deployment on an edge device. We perform transfer learning on a pre-trained MobileNetV2 using a dataset of 800 images (400 AP, 400 LA). We employ 5-fold cross-validation, where each fold has 640 training images and 160 test images. Subsequently, we apply multiple quantization techniques—including FP32, FP16, dynamic, and Int8 to reduce the model's size and enhance inference speed. Evaluating across the 5 folds (160 test images per fold), our evaluation showed that quantization preserves over 98% classification accuracy while reducing the original model size by over 4x for the quantized variants. We compare the performance on a personal computer (PC) with a graphical processing unit (GPU) and on an edge device. Although the GPU-based implementation exhibited lower warmup and steady-state inference times, the steady-state performance on the edge device remains competitive despite higher initialization overhead. Our results show that we can use transfer learning to leverage large-scale pre-trained models for specific applications. Our quantization strategies enable efficient, real-time AP/LA X-ray view classification on an edge device, making it a promising solution for clinical use.

**Keywords:** Transfer Learning, Model Quantization, Edge Computing, Medical Image Classification

## 1. Introduction

X-ray imaging is among the most frequently performed examinations in medicine, with billions of radiographs obtained annually (Akhter et al., 2023). Clinical protocols often require multiple views of the same region. For example, a standard X-ray exam typically includes a frontal (anteroposterior, AP, or posteroanterior, PA) and a lateral (LA) projection (Fang et al., 2021). Correctly identifying the view orientation (AP vs LA) is crucial for accurate interpretation, as anatomical structures appear differently across views (Baltruschat et al., 2019). Xue et al. used a traditional machine learning pipeline for AP/LA classification, involving image pre-processing, hand-crafted features, followed by classification using a Support Vector Machine (SVM) (Xue et al., 2015). Mitsuyama et al. proposed a deep learning approach using EfficientNet models to classify chest X-rays by projection (AP, PA, lateral) and rotation (upright, inverted, left/right), trained on large multi-institutional datasets (Mitsuyama et al., 2025). However, these methods do not specifically address real-time inference on resource-limited edge devices.

Building upon these methods and addressing the need for efficient inference on resource-constrained devices, we propose a framework that uses both transfer learning and model

quantization, specifically for deployment on a small form factor and resource-constrained edge device. We utilize a MobileNetV2 backbone pre-trained on large-scale natural image datasets to efficiently capture discriminative visual features relevant for distinguishing AP from LA X-ray views. Subsequently, we apply multiple quantization strategies to reduce the model's size and computational demands without compromising accuracy. By optimizing for real-time inference on an edge device, our method addresses clinical needs for both rapid image analysis and accurate classification in settings with limited computational resources.

## 2. Methods

The study uses a dataset of 800 X-ray images obtained from the BLUU dataset (Faculty of Informatics, 2021). The images are evenly split between anterior-posterior (AP) and lateral (LA) views, resulting in 400 samples per view. We performed 5-fold cross-validation, splitting the dataset into 5 folds of equal size (160 images per fold), preserving a 20% test-train split within each fold. Preprocessing steps included resizing images to 224×224 pixels and normalizing pixel values by rescaling them by a factor of 1/255. Horizontal flipping was applied as a data augmentation strategy for the training set.

MobileNetV2 was used as the backbone for our classification model due to its lightweight architecture (Sandler et al., 2018). The network was pre-trained on ImageNet (Deng et al., 2009), and its top layers were removed. A custom classification head was added consisting of a Global Average Pooling layer, a dense layer with 128 units and ReLU activation, and a final dense layer with 2 units using softmax activation for binary classification. In this transfer learning approach, the pre-trained MobileNetV2 base was frozen to preserve its learned features while training a new classification head on the AP/LA training data.[1]

To optimize deployment on an edge device, we converted the trained model to TFLite format, yielding a baseline floating point 32-bit (FP32) model. We then applied three quantization techniques (a) **FP16**: Uses 16-bit floats for weights and activations, nearly halving model size versus FP32 while preserving accuracy, (b) **Dynamic**: Pre-quantizes weights to 8-bit integer (Int8) while converting activations on the fly during inference, and (c) **Int8**: Quantizes weights to Int8 but retains float precision for the input/output layers (Rokh et al., 2023). Each variant was generated using `TFLiteConverter.from_keras_model(model)` and evaluated for model size, inference latency, and classification accuracy.

The performance metrics used are model size in kilobytes (KB), inference latency in milliseconds (ms) and classification accuracy percentage. All experiments were implemented in Python using TensorFlow and Keras. Data preprocessing and augmentation were handled via `ImageDataGenerator`. For each fold, training was conducted with a batch size of 32 for 5 epochs using the Adam optimizer (learning rate = 1e-3). After training on four folds (640 images), we tested on the remaining fold (160 images). To obtain final performance metrics, we repeated this process for all the folds and averaged the results. The final models were saved in Keras and TFLite formats. Performance comparisons were conducted on a system equipped with an NVIDIA RTX 4080 GPU and an edge device (Jetson Orin Nano Super 8 GB). File and directory operations, including dataset splitting and model export, were managed using Python's `os` and `shutil` modules.

---

1. This avoids over-fitting on 640 training images, reuses ImageNet features, and cuts training time.

## 3. Results and Discussion

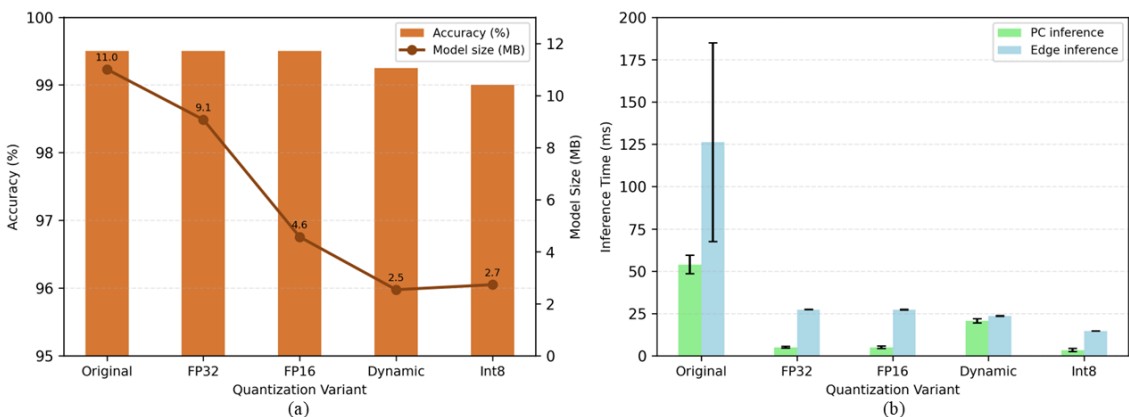

Figure 1: (a) Comparison of model size and classification accuracy across different variants, measured on a GPU-equipped PC and a Jetson edge device. The accuracy remains consistently over 98%. (b) Average inference time per sample for each quantization variant, highlighting differences between PC and edge device performance.

Figure 1 shows the results of the classification accuracy versus the model size, in addition to a comparison of the inference time on the PC (GPU) versus the Edge device (Jetson). The quantized models achieved near-perfect classification accuracy—with FP32, FP16, and the original model all reaching an average classification accuracy of 99.5%, while the dynamic and Int8 variants showed slight reductions (around 99.25% and 99.0%, respectively).

Model quantization reduced the storage footprint compared to the original model, from 11,273 KB down to as low as 2603 KB (for dynamic variant) and 4,677 KB (for FP16). The quantized model sizes on the Jetson closely match those on the PC, with only minor variations due to file system metadata and serialization. On the PC, steady-state inference latency is lower. For instance, the Int8 variant achieves around 3.43 ms compared to 14.66 ms on the Jetson, with the original model achieving roughly 54 ms on GPU versus 126 ms on Jetson. These differences demonstrate the superior hardware acceleration on the GPU, although the Jetson's steady-state performance remains competitive for edge applications despite its higher latency. The warmup latency is dramatically higher than the steady-state latency, often 20 to 30 times slower due to one-time overheads such as model initialization and memory allocation. Once these processes are completed, the model achieves much faster and consistent inference times.

Future work will expand the training corpus beyond frontal-lateral chest radiographs to include posterior-anterior (PA) as a distinct class and incorporate other modalities such as CT and MRI, allowing the model to handle a wider range of anatomical views and imaging techniques. Moreover, recent advances (Tian et al., 2023; Bimbraw et al., 2024; Van et al., 2024) demonstrate that large vision-language models hold significant promise for medical image understanding and classification. Deploying these optimized models on edge devices has the potential to support clinical workflows by enabling efficient, real-time X-ray view classification in settings with limited computational resources.

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
