# OpenReview forum: "Transfer Learning and Quantization for Efficient AP vs. LA X‑Ray View Classification on an Edge Device"
_MIDL.io/2025/Short_Papers — MIDL 2025 - Short Papers_

### Official Review · Reviewer_HTkH · 2025-04-23

**Rating:** 3
**Confidence:** 4

**Summary:**

The authors propose a AP vs. LA X-ray view classifier using a MovileNetV2 network pretrained on ImageNet weights and investigate three quantization techniques for deploying the model on an edge device. They use a dataset with 800 X-ray images, perform 5-fold cross-validation, and report several metrics: model size, inference latency and classification accuracy.

**Strengths:**

- This paper focuses on efficient AP vs. LA X-ray view classification with limited compute
- The authors perform 5-fold cross-validation
- Three quantization techniques were evaluated
- Several metrics were reported: model size, inference latency and classification accuracy
- The authors discuss some ideas for future work.

**Weaknesses:**

- The abstract includes too much experimental details.
- What about the distinction between AP and PA classification? I believe this difference is also crucial for the accurate interpretation of anatomical structures, which appear differently across views. While this is mentioned as a limitation, the AP/LA task classification doesn’t seem particularly challenging.
- The authors do not provide a justification for the decision to freeze and retain the pretrained features.
- The graph in Figure 1 (a) is difficult to interpret. I recommend plotting accuracy on the y-axis and using shape size or a similar method to represent the model size.

---

### Decision · Program_Chairs · 2025-05-01

Accept